# Reflections on Early Stages of Environmentally Assisted Cracking from Corrosion Pits

**Alan Turnbull**

National Physical Laboratory, Teddington TW11 0LW, UK; alan.turnbull@npl.co.uk

**Abstract:** A perspective is presented on the evolution of damage due to environmentally assisted cracking (EAC), from crack precursor development through to long crack growth. The variable nature of crack precursors is highlighted with an observation that uncontrolled chemistry excursions or fabrication defects could eliminate any significant delay associated with that step in the damage evolution process. Specimen preparation by machining and grinding can be critical in determining the apparent susceptibility of the metal to EAC and corrosion, and an example for 316L stainless steel is given to show how physical defects generated by the grinding wheel can become the dominant site for pitting attack relative to MnS inclusions. Corrosion pits are the most commonly observed precursor to cracks in aqueous chloride environments. The loci of sites of crack initiation around a pit are discussed and the inherent challenges in quantifying the growth of cracks smaller than the pit depth described with implications for modelling of the pit-to-crack transition. The remarkably enhanced stress corrosion crack growth rate data for short and small cracks in a 12Cr steam turbine blade in a simulated condensate environment are discussed in the context of crack electrochemistry modelling and the implications for engineering integrity.

**Keywords:** stress corrosion cracking; corrosion fatigue; pitting; surface finish; modelling

## 1. Introduction

The evolution of damage development in engineering components and structures is often characterised schematically in diagrams such as that of Figure 1. One can find a similar depiction by Staehle for example [1]. These diagrams are of most value for applications where crack growth can be tolerated by virtue of accessible inspection and relatively slow crack growth rates. In applications where inspection is not feasible, or crack growth rate too rapid, the emphasis is on ensuring no damage development beyond Stage 2, i.e., any crack, should it form, would be non-propagating. The approach in that case is either to select materials resistant to cracking, design below the threshold stress or threshold stress amplitude for sustained cracking, or to manage the operating system to minimise the likelihood of cracking, for example, by water chemistry control.

These diagrams describing crack size evolution can be somewhat misleading in not accounting explicitly for the distributed nature of flaw sizes (with an inherent uncertainty in the possible evolved value of crack length at any time). Additionally, in many applications, the relative time-domain allotted to the different stages can be markedly different from those suggested. They also give the impression of a continuous uninterrupted crack growth process, whereas, in many service applications, the crack growth may be intermittent, with occasional crack arrest, and dependent on controlled or uncontrolled excursions in environment or stress (e.g., scheduled outages or temporary loss of control of water chemistry or temperature). Accordingly, the assumption of a significant fraction of life in precursor development prior to crack formation may be obviated by a major excursion, in plant water chemistry for example.

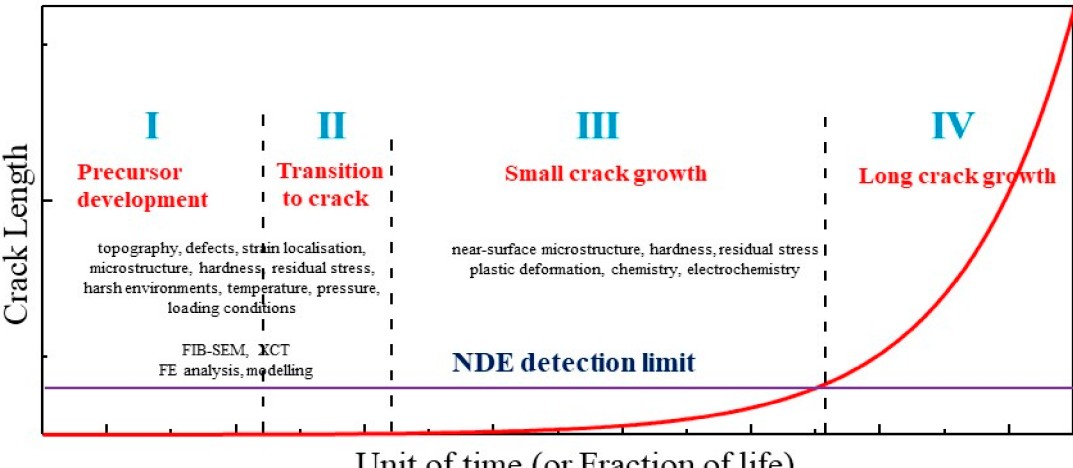

**Figure 1.** Schematic depiction of the evolution of crack size as interlinked stages of development.

Despite these limitations, the virtue of such diagrams is that they highlight the stages of damage development that require quantification if the time to the crack detection limit and life prediction are to be estimated with any measure of confidence. They also indicate where research and development in materials, fabrication, and system operation should be targeted to achieve the goal of increasingly longer lives. In the latter respect, Staehle's focus [1] on improved characterisation of the processes determining precursor development at the nano-to micro-scale points the way forward.

The nature of the crack precursor depends on the material-environment system. Examples from nuclear applications have been described recently by Persuad et al. [2] and by Bruemmer et al. [3]. Here, examples from our own work in Figure 2 highlight the varied nature of crack precursors. These include corrosion pits formed on a 12Cr martensitic stainless steel (MSS) in chloride environments [4]; dealloyed layer formed on a duplex stainless steel (DSS) in evaporating seawater conditions at elevated temperature (the dealloying attributed primarily to a nanocrystalline layer induced by surface grinding) [5]; weld defects, in this example for a weldable 13Cr MSS in a simulated oilfield environment. [6].

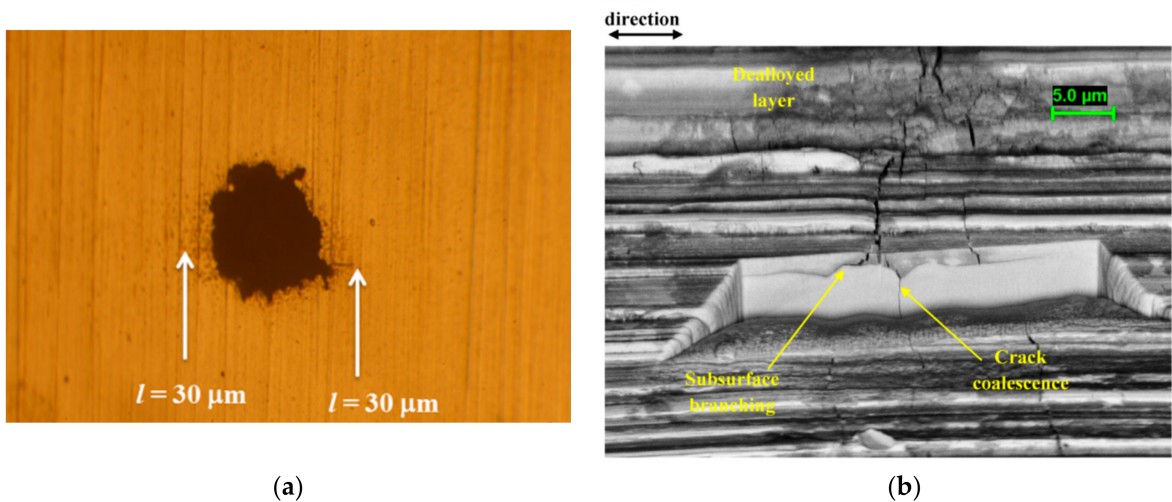

(a)          (b)

**Figure 2.** *Cont.*

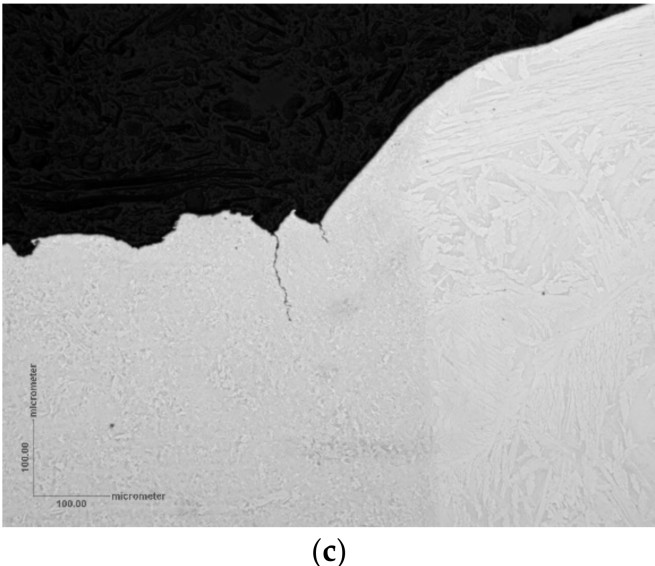

(c)

**Figure 2.** Examples of crack precursors; (**a**) cracks emerging from corrosion pit in 12Cr blade steel in simulated steam turbine condensate. Reprinted with permission from [4] Copyright @ National Physical Laboratory; (**b**) stress corrosion cracks formed from dealloyed layer in DSS in evaporated seawater conditions at elevated temperature. Reprinted with permission from [5] Copyright 2016 Crown Copyright; (**c**) stress corrosion crack in welded 13Cr MSS in simulated oilfield brine in $H_2S/CO_2$ at elevated temperature [6]. Figure 2c is copyright © Institute of Materials, Minerals and Mining, reprinted by permission of Informa UK Limited, trading as Taylor & Francis Group, on behalf of Institute of Materials, Minerals and Mining.

The sharp triangular-shaped defects in the welded 13Cr MSS highlight an application where the precursor development stage may be effectively eliminated as a consequence of the fabrication process.

While there are different forms of crack precursor, depending on the application, this brief overview/commentary will focus on corrosion pits, as perhaps the most commonly investigated. After drawing attention to the importance of surface finish to pit development in stainless steels, the focus is on the more generic aspects of the pit-to-crack transition and the challenges of small crack growth rate measurement from pits. In-house data and modelling for small, short and long stress corrosion crack growth rates in 12Cr steam turbine blades are then used to highlight the enhanced growth rate that can be associated with small cracks developed from pits. Finally, completing the stages represented in Figure 1, the implication for turbine life is briefly discussed.

## 2. Impact of Surface Preparation

The surface preparation method and the final finish adopted for laboratory test specimens represent a potential source of disparity in laboratory test results and also their relevance to service. Standard tests for pitting and for EAC typically recommend that specimens should be prepared to a finish that reflects that in service or specify that specimens should be ground to a maximum average surface roughness, Ra. The latter approach is the more commonly adopted but it has two limitations. Specimens may be ground to the desired surface finish but if the grinding is not carried out in a progressive way, i.e., with the previous damaged layer removed during each stage, sub-surface damage from prior machining may remain. The second limitation is that engineering workshops will tend to grind EAC test specimens with a grinding wheel, almost inevitable for cylindrical specimens, and this impacts on surface properties quite differently from hand-grinding. In some cases, the latter might be followed by polishing or electrochemical polishing.

Grinding with a grinding wheel can give rise to near-surface residual stress, the introduction of physical defects, plastic deformation, the formation of a nanocrystalline layer,

and high local hardness, the extent of these being dependent on the relevant alloy [5,7–12]. The consequence is that the material actually exposed to the environment may be quite different from the bulk alloy with respect to both microstructure and mechanical properties. Superimposed on this modified near-surface material may be an array of physical defects formed from the grinding wheel process associated with chipping, gouging and deep grooving.

The importance of physical defects in acting as a precursor to pitting corrosion is exemplified for 316L stainless steel (SS) in simulated oilfield environment containing $CO_2$ and $H_2S$ at elevated temperature in Figure 3 [9]. Here, tests were carried out with and without heat-tinting. Prior to testing, the location of inclusions and physical defects on the surface was mapped at high resolution using a scanning electron microscope, and, post-test, the number of pits and their location with respect to inclusions and pre-existing defects were assessed. Notably, there was not a simple correlation with Ra value, which perhaps reflects the frequency of dressing of the grinding wheel. The key observation is that for most cases there was a greater number of pits associated with physical defects compared to inclusions. This raises the question as to the perceived role of inclusions as the commonly assumed primary source of pitting in stainless steels. Is this observation simply a product of the fine surface finish in most laboratory tests, with specimens usually hand-ground or more finely polished? As such, is the adoption of an ideal surface finish in laboratory testing producing results that are specific only to that testing and perhaps misleading when extended to real components in service? More testing with a surface finish representative of that adopted in service is recommended. This does not preclude the role of chemical inhomogeneities acting as pit precursors but ensures that their significance is appropriately weighted.

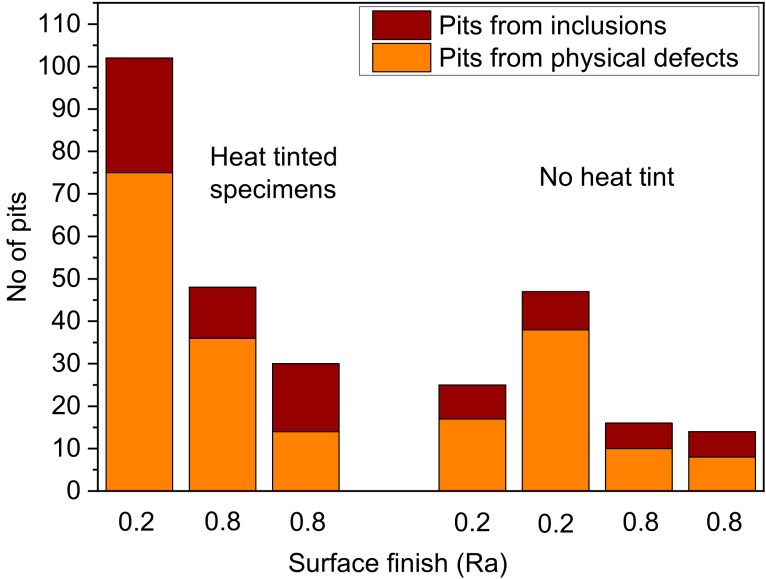

**Figure 3.** Comparison of number of pits associated with inclusions relative to physical defects in 316L SS in simulated oilfield environments. Reprinted with permission from [9] Copyright 2013 Crown Copyright.

## 3. Pits and the Pit-to-Crack Transition

Traditionally, the role of pits as effective precursors to cracks has been ascribed to a combination of stress/strain concentration, an aggressive local environment, and a film-free active surface. Turnbull et al. [13] proposed an additional concept impacting the evolution of cracks from pits; specifically, dynamic straining induced by the pit growing in its local strain field. Finite element analysis (FEA) supported that concept and predicted plastic strain rates in the range associated with stress corrosion cracking in slow strain rate testing.

In the FEA study, undertaken for a 3NiVrMoV steam turbine disc steel, plastic strain was concentrated just below the pit mouth (Figure 4). This correlated with the location of cracks emerging from pits, as shown by detailed X-ray computed tomography, XCT [14].

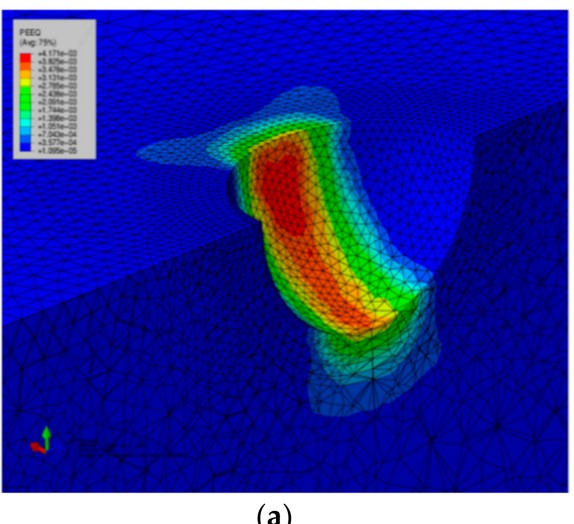

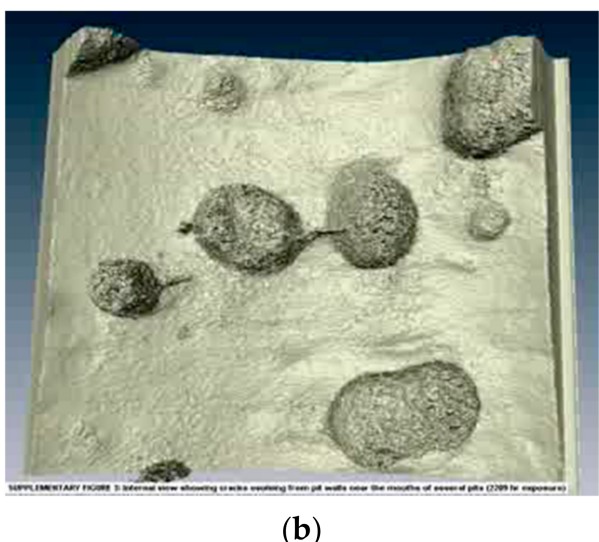

(**a**)  (**b**)

**Figure 4.** (**a**) FEA modelling of strain localisation in pit. Reprinted with permission from [13] Copyright 2010 Crown Copyright; (**b**) 3D imaging of cracks emerging from pit near the pit mouth using XCT [14]. Figure 4b is copyright Elsevier, 2011, reproduced by permission.

The XCT observation, with the support from FEA, represented a step-change in thinking about the evolution of cracks from pits. Further work on fatigue crack development from pits in a 12Cr MSS supported the idea that cracks initiated near surface rather than at the pit base [15]. Indeed, even for shot-peened 12Cr MSS, with a compressive residual stress gradient, fatigue cracks from pits of depth 120 µm or less all initiated away from the base (fatigue tested in air) [16]. In the latter case, this was attributed to sub-surface deformation rather than the stress or strain profile. These observations do not preclude the possibility that cracks may initiate at, or very close to the pit base, as suggested for fatigue crack development from pits in Al alloys as one example [17]. What it does highlight is the need to abandon preconceived ideas about where a crack may evolve from a pit. The specific location will depend on pit geometry, which determines how macro-stresses and macro-strains are distributed, micro-topographical features, and near-surface deformation.

An interesting feature of the research of Guo et al. [16] was the observation of multiple small fatigue cracks emerging from the pit at different depths from the surface (Figure 5). As noted by Burns et al. [18], micro-topographical features offer a local stress/strain condition and so multiple cracks are not unexpected. However, most will become non-propagating, with those that survive and link up to form a fully developed crack around the pit being most influenced by the macro-stress/strain field and favourable microstructure.

It is stating the obvious that a crack emerges from a pit with a size that is smaller than the pit size. However, the point is stressed because of the challenge this poses in quantifying the crack growth of such cracks and the growth rate at the pit-to-crack transition when these cracks first emerge. Optical techniques based on microscopic viewing of the surface crack image only the crack emerging at the surface and this may be some time after initiation. Additionally, corrosion product formation can readily obscure the crack. Digital image correlation, notwithstanding similar limitations associated with corrosion product, detects that there is a change in surface strain associated with crack development, but this could relate to a single crack or multiple cracks. The same limitation applies to the potential drop method. A change in potential drop signifies a change in resistance but this could be due to a single crack or two, with no distinction, and may be affected by growth of the pit itself. The latter would apply also to digital image correlation. In the absence of any

pit growth, both digital image correlation and potential drop would indicate that cracking is developing, so can give some measure of the time to initiation, within the constraint of measurement resolution, if not the actual growth rate.

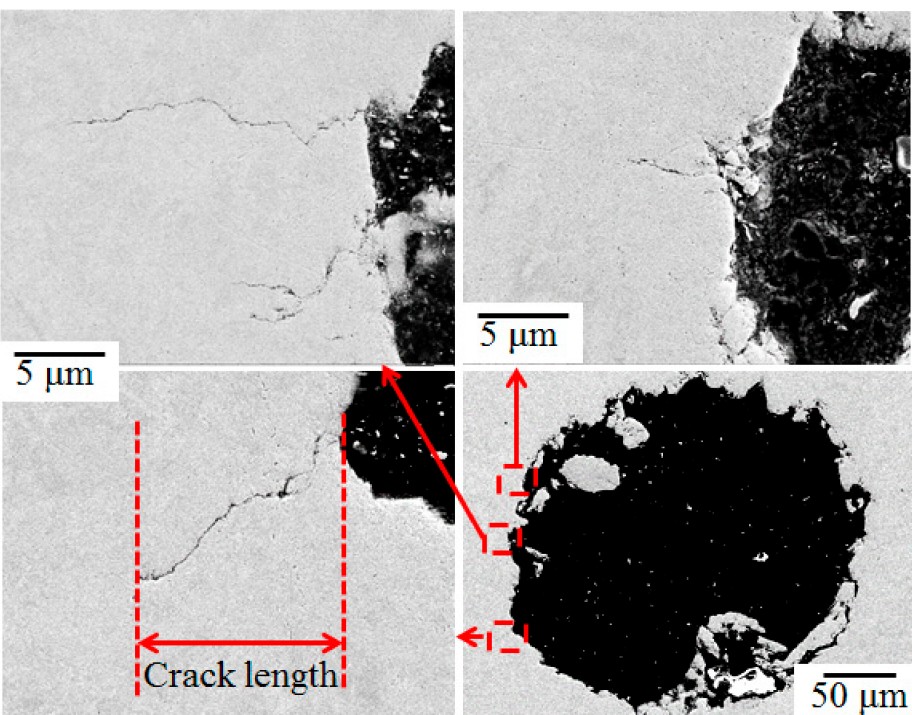

**Figure 5.** SEM images of fatigue microcracks emerging from a pit of 209 μm depth in a 12Cr MSS. The layer removal method was used to characterise crack development as a function of distance from the surface; the cracks shown are after a layer removal of 51 μm. Reprinted with permission from [16] Copyright 2017 Crown Copyright.

The upshot is that we will not know what the crack growth rate is at the pit to crack transition, yet that features in the classic work of Kondo [19] that is often adopted in models of pitting and cracking. Kondo proposed the phenomenological and necessary criteria that the pit must be greater than a certain size and that the crack growth rate is greater than the pit growth rate. It should be emphasised that the threshold pit size is related to a threshold mechanics criterion and by implication will depend on the stress and strain distribution around the pit; thus, pit geometry as well as depth could be factor. Predicting the threshold pit size is challenging and experimental observation is usually required (see Reference [20] as example). The second criterion makes sense because the crack has to "outrun" the pit to survive and this explains why small slow growing pits above a critical size may develop cracks but deeper pits, that were growing fast by implication at the time of inspection, do not exhibit cracking. In the special case of high-frequency fatigue testing, the fatigue crack growth rate, da/dt, is usually greater than the pit growth rate and the transition to a crack occurs as soon as the threshold pit size is achieved.

In applying his concept to corrosion fatigue, Kondo used the crack growth rate from a separate short crack growth rate test. The reality is that we do not know whether the short crack growth rate measured in fracture mechanics testing bears a close relationship to the actual growth rate of the small crack developing from the pit at the pit-to-crack transition. These small cracks would be of the scale of the microstructure which has implications not only for growth rate but also for assigning a stress intensity factor for such cracks. By default, Kondo implicitly assumes that the crack is the same depth as the pit at the transition and this is clearly not correct but from a modelling perspective it is a very convenient assumption.

Assigning a stress intensity factor, K, to cracks smaller than the pit is inherently inexact. Murakami and Endo [21] developed the concept of a projected pit and crack area, with a crack on either side of the pit, perpendicular to the stress axis, and used the square root of the projected area in the definition of K for that case. However, FEA [22] has shown that a crack on one side of a pit (relative to stress axis) is mechanically independent of a crack on the other side until both cracks approach the depth of the pit, beyond which they form a continuous crack front. Furthermore, in most applications, plasticity develops around the pit. In applying linear elastic fracture mechanics there is no limit to the stress concentration that may develop as there is no constraint associated with plasticity (Figure 6). This is well known but tends to be overlooked.

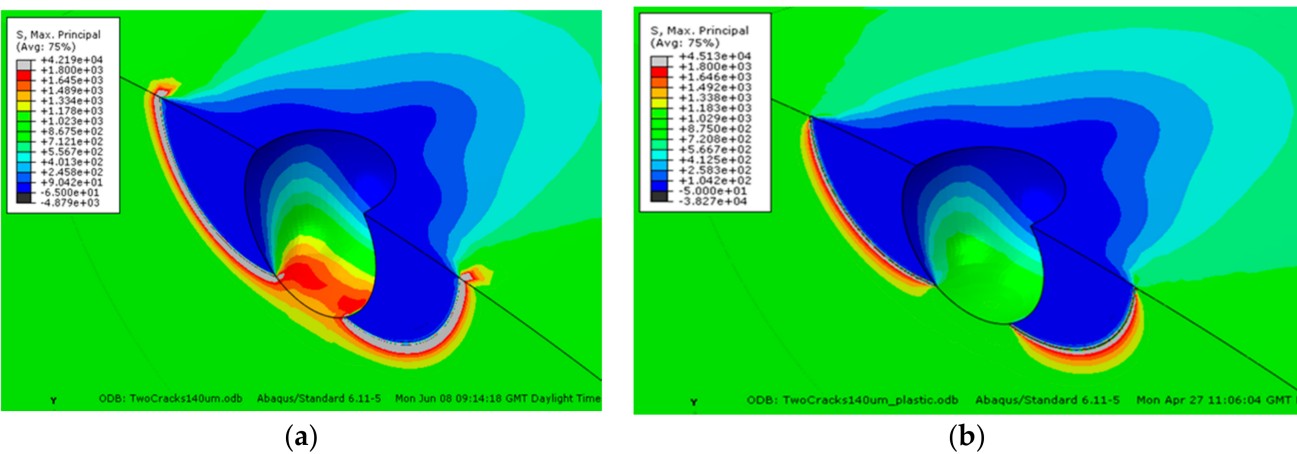

(**a**)                                    (**b**)

**Figure 6.** Stress distribution (in MPa) around 150 μm deep pit and 140 μm deep crack calculated assuming (**a**) pure elastic conditions, (**b**) elastic-plastic conditions. Remote stress applied is 90% $\sigma_{0.2}$ [22].

Hence, there is inherent uncertainty in crack size measurement for cracks smaller than the pit depth and uncertainty over applicability of K. The key conclusion is that only when the crack is fully developed beyond the pit depth, with a continuous crack front, and unaffected by the plastic strain profile of the pit can a reliable basis for crack growth rates as a function of crack size and K be obtained. Such a crack is exemplified schematically in Figure 7, extracted from the FEA [22], showing a fully developed crack that extends beyond the pit depth and in this case beyond the plastic strain field of the pit.

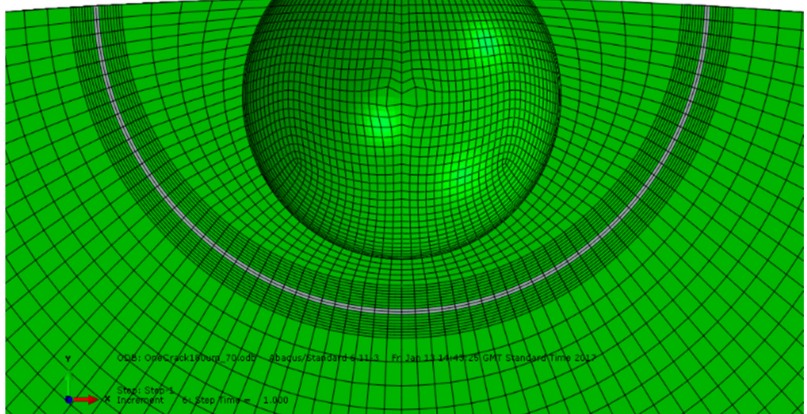

**Figure 7.** Illustrative FEA mesh around a pit to highlight a continuous crack front extending beyond the pit depth and in this case the crack was just outside the field of plasticity associated with the pit, enabling assignment of K value [22].

Clearly, if the goal is to measure crack growth rates for very small cracks growing from pits, the pit should be as shallow in depth as possible within the constraints of the threshold size for crack development being exceeded and the absence of other competing defects.

## 4. Test Methodology for Quantification of Small Crack Growth Rate from Pits

With the exception of very specific applications for which crack-front marking at different times is feasible, as in fatigue of aluminium alloys, the most effective test methodology for measuring crack growth from pits is based primarily on satisfying the conditions related to the concept above and Figure 7. More guidance can be found in ISO 21153 [23]. Specifically, develop a single pre-pit to the required depth using an appropriate method such as the galvanostatic droplet method [23]; ideally, fatigue pre-crack until the crack depth is judged to be just greater than the maximum depth of the pit. The fatigue pre-cracking gives a crack-front position for the initial crack size before the onset of the stress corrosion or corrosion fatigue test. Monitor the subsequent crack extension under the applied static or dynamic loading and environmental exposure conditions by an appropriate method, such as the direct current potential drop (DCPD) technique. The probes of the latter should be isolated and positioned on either side of the pre-pit with the probe spacing a balance between signal resolution and minimisation of interference of the corrosion process local to the pre-pit.

## 5. Pitting and Cracking of Steam Turbine Blades

### 5.1. Fatigue

Steam turbine blades tend to have relatively long lives but when failure does occur, it is most often the result of pitting followed by high-frequency fatigue, predominantly in the low-pressure turbine stage, for which stresses are higher and condensation is more prevalent. There are now guidelines for assessing the likelihood of blade failure by high cycle fatigue based on the following steps [24]: determine experimentally the threshold pit size for fatigue cracking as a function of stress amplitude and stress ratio, expressed in a Kitigawa–Takahashi plot; undertake stress analysis of blades noting that stress amplitudes are designed to be small and the stress ratio is large; measure the distribution of pit sizes in service during inspection. A decision can then be made based on measured pit sizes as to whether to remove the blades for repair or to defer until the next inspection.

This is a satisfactory approach for base-loading where the turbines are essentially in continuous use, as in nuclear power plants. However, there has been increasing adoption of two-shifting in fossil fuel power plants for economic reasons, or to balance out fluctuations in renewable electricity supply in the national grid. In two-shifting, the power plant is operational for approximately 16 h on-load per day and 8 h off-load, with shut-down often at the weekend. Since the start-up involves a 20 min rise time (hot start), with the stress ranging from effectively zero to 90% of $\sigma_{0.2}$ in a cycle, this introduces the possibility of low-frequency corrosion fatigue and this was an initial focus for research [25]. However, while corrosion fatigue cyclic crack growth rates were high, the number of cycles per annum is inherently low. Hence, while not negligible, the impact of low-frequency corrosion fatigue on remanent life would be projected to be small.

### 5.2. Stress Corrosion Cracking

There is an incidental effect of two-shifting, which has been generally overlooked. On-load, the turbine chamber becomes deaerated. Off-load, it is effectively aerated. It can take a significant period for the large chamber to attain a low oxygen level on-load. In addition, the time evolution of the electrode potential tends to lag the decreasing oxygen level. Thus, during the 16 h period of maximum stress, the chamber in the low-pressure stage must be treated as if it were fully aerated, with approximately 1.8 ppm oxygen at a temperature of 90 °C, the temperature typical of first condensation in the low-pressure turbine stage. With aeration, stress corrosion cracking becomes a potential failure mechanism. Under normal water chemistry conditions, the chemistry of the liquid film formed on the blades

can be represented typically by a solution of 300 ppb (by mass) chloride ion and 300 ppb sulphate ions, assuming 1% wetness [26]. This anion concentration is too low to cause stress corrosion cracking from a corrosion pit. However, excursions in water chemistry do occur in operating plant. This leads to the question as to what stress corrosion crack growth rates are feasible and what are the implications for existing inspection protocols.

Inspection intervals for steam turbine power plant are based on long crack growth rate data and the detectable crack length may be as much as 3 mm (due to the complex geometry and accessibility to inspection tools, cracks shorter than this could conceivably go undetected). The challenge is to quantify the period between a crack developing from a pit and the evolution to a detectable length.

To address that challenge, stress corrosion crack growth rate measurement on FV566 12Cr MSS was undertaken [25] based initially on short crack growth rate measurement using fracture mechanics specimens (single edge notched tensile, SENT), to correlate with the modelling described below, and small crack growth rate measurement from a pit and pre-crack following the methodology described in Section 4. (A short crack refers to a through-thickness crack in a fracture mechanics specimen that is initially small in two dimensions but long in one (through thickness); a small crack refers to a crack that is initially small in three dimensions (in particular, both length and depth) in comparison to a relevant microstructural scale, continuum mechanics scale or physical size scale [23].) Long crack growth rate measurements in this investigation refer to cracks in compact tension (CT) specimens with length in excess of 6 mm. This FV566 12Cr MSS contains approximately 1.7% Mo so has reasonable resistance to pitting at 90 °C, with a critical chloride ion concentration for pitting between 500 and 1000 ppm. The observation of pitting in service suggests that this critical concentration is exceeded at some time. However, such a large excursion is a comparatively rare event and would be considered atypical. In the latter context, guided by UK industry, a chloride ion concentration of 35 ppm was adopted as being a better reference concentration for first testing the impact of chemistry excursions on stress corrosion crack growth rates.

The measured crack growth rates are plotted in Figure 8 as a function of crack size, rather than the usual da/dt vs. K, which does show the conventional plateau region [26] but is less informative. Testing was carried out at 90 °C and the corrosion potential was approximately −0.15 V SCE for all tests.

The threshold K value for sustained crack growth was approximately 16 MPa m$^{1/2}$. Two tests showed a significant crack growth rate for the short crack but the cracks appeared to arrest. For these tests, fractography indicated only a couple of isolated regions of intergranular cracking along the crack front. The K value was apparently insufficient to activate adjacent less susceptible grain boundaries and form a continuous crack front. The growth of the intergranular regions was then pinned by those less susceptible regions. For the tests showing sustained crack growth, the short crack growth rate in the SENT specimens was over 20-fold greater than that for long cracks, up to a crack length of approximately 1.5 mm, with cracking wholly intergranular along the crack front. The results pose the question as to when does a short crack become a long crack. The particular SENT specimens had limitations in extending to longer cracks. Accordingly, a CT specimen, fatigue pre-cracked to approximately 1.3 mm below the notch, was adopted to address that question. The growth rate from the CT crack approached that for the short crack in the SENT specimen, which is reassuring, but fell rapidly towards that associated with the long crack.

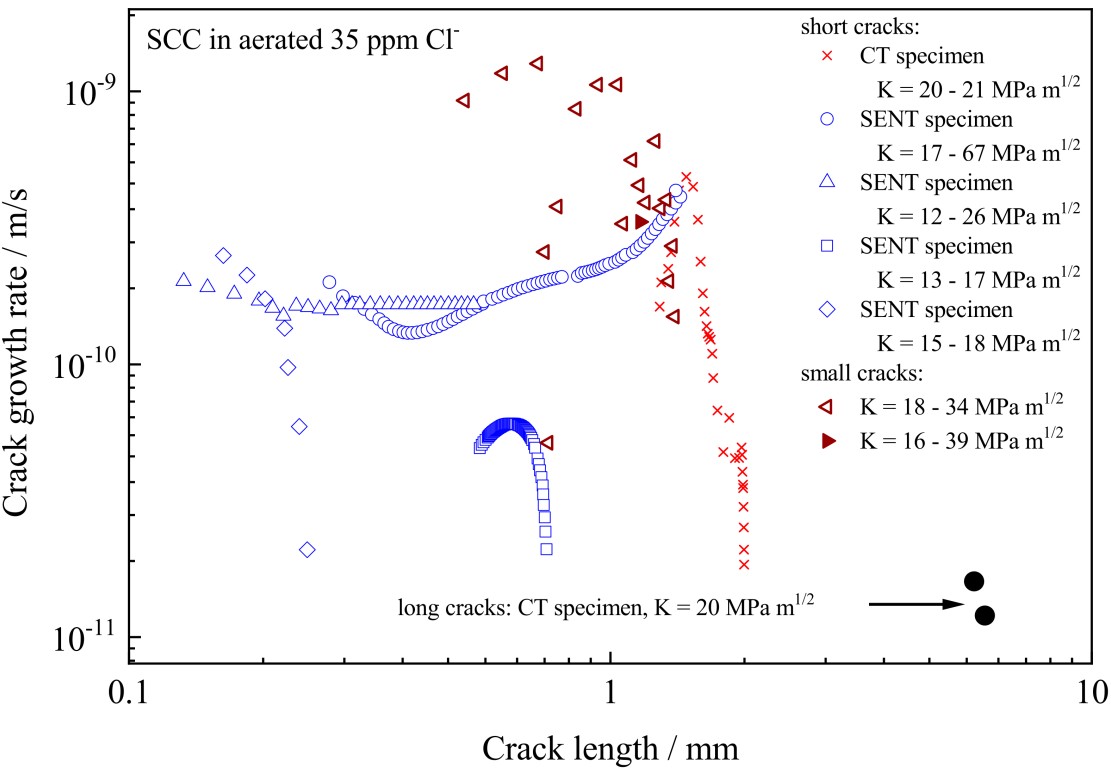

**Figure 8.** Stress corrosion crack growth rates for FV566 12Cr MSS in aerated 35 ppm Cl⁻ at 90 °C [26]. Reproduced by permission from De Gruyter Publishing.

Testing with small cracks grown from pits and a fatigue pre-crack, although limited in the number of tests undertaken, showed even higher crack growth rates, albeit the growth rate was more variable. In one test, the potential drop probes failed, so only a time-averaged crack growth rate is quoted at an average crack depth.

*5.3. Modelling the Electrochemical Crack Size Effect*

A crack size effect on growth rate in aqueous solution, unrelated to mechanical effects, was not unexpected. The concept of an electrochemical crack size effect on crack growth rate was first proposed by Gangloff [27] for a high-strength steel in simulated marine environments and supported by mass transport modelling of crack electrochemistry [28]. In that application, the combination of a change in potential drop with crack size and critically an acidification of the short crack provided an explanation. For this steam turbine application, the distinguishing feature is the low solution conductivity. It would be anticipated conceptually that significant changes in crack-tip potential between short and long cracks would arise, simply for that reason. Predictions of electrochemical modelling [29], assuming a short crack in a fracture mechanics specimen (modelling small crack electrochemistry with the extra dimension is a bigger challenge), supported that expectation (Figure 9) with the crack-tip potential decreasing with crack length, most significantly for the lowest solution conductivity. However, it is notable that the concentration of both chloride ions and hydrogen ions at the crack-tip increases with increasing crack length, suggesting that the chemistry would be more aggressive for the longer crack. In the model, the current density on the crack walls is assumed to be associated with the passive current. However, as noted previously in the text, pitting of this FV566 12Cr steel occurs between 500 and 1000 ppm chloride and this would render that assumption unreasonable at crack length greater than 3 mm for the 35 ppm bulk chloride case. In testing, there was no specific evidence of pitting attack on the fracture surface, but an oxide film was formed that was

challenging to remove by cleaning. The author appreciates the comment from Gerald Frankel on this issue).

The challenge is to convert this chemical and electrochemical information into crack growth rate prediction. With very few exceptions we do not have robust models capable of quantitative prediction of crack growth rates that incorporate crack-tip electrochemistry, crack-tip reaction process and crack-tip strain rates, not least because of the lack of data for the functional dependence of crack-tip reaction kinetics of a exposed grain boundary on local chemistry and potential. The best that can be achieved is to draw inferences for the trends in crack growth rate using modelling to guide interpretation of test data.

In that context, the test data at a bulk concentration of 35 ppm of chloride ions show an increased growth rate for short cracks that is consistent with the more noble potential for the short crack, despite an increased pH and reduced concentration of chloride ion. Additionally, the crack-tip potential for all crack lengths is quite noble at this bulk concentration of chloride ions. These two observations rule out any stress corrosion cracking mechanism related to hydrogen and suggest a mechanism associated with anodic reaction. A perfect correlation between the modelling predictions at constant K (as adopted in the modelling to best highlight the crack size effect) with the test data for which K is increasing would not be expected. An increase in K will mean greater crack opening and a reduction in potential drop with crack length. When combined with changes in chemistry, it should not be expected that the crack growth rate dependence on crack length will mimic the smooth change in potential in Figure 9b. Nevertheless, the apparent dramatic decrease in crack growth rate observed for the compact tension test is very hard to reconcile with modelling predictions.

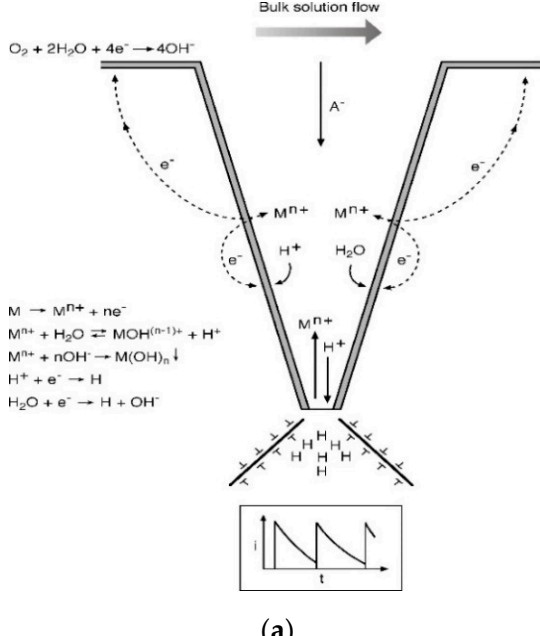

(**a**)

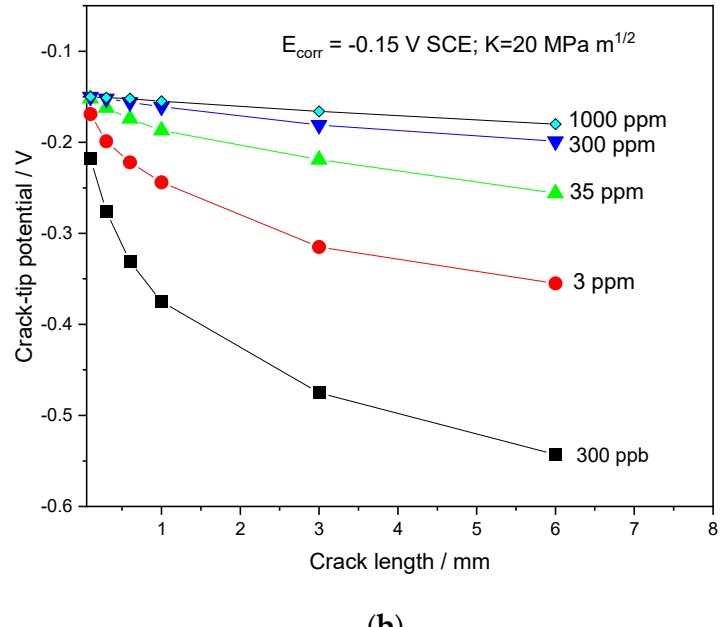

(**b**)

**Figure 9.** *Cont.*

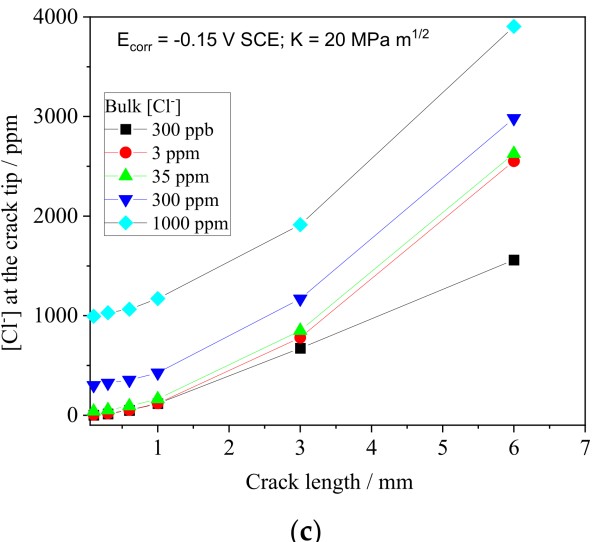

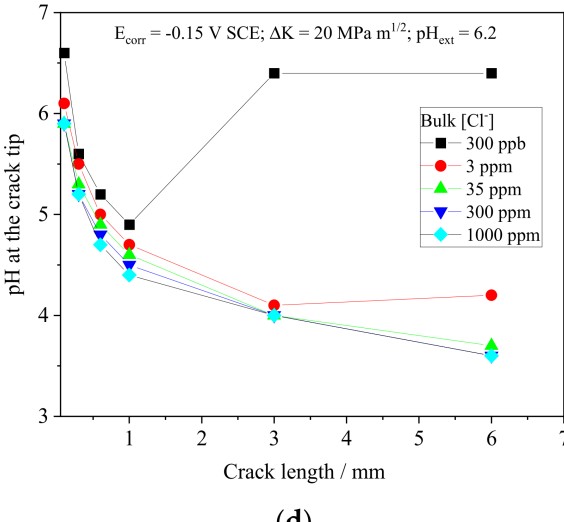

**Figure 9.** Modelling of crack-tip electrochemistry for FV566 12Cr MSS in chloride solution of low conductivity, reprinted with permission from [29] Copyright 2017 Crown Copyright: (**a**) schematic of processes involved; (**b**) dependence of crack-tip potential on crack length and chloride ion concentration; (**c**) crack-tip chloride ion concentration; (**d**) crack-tip pH.

### 5.4. Implication of Measured Crack Growth Rates for Turbine Life

The high crack growth rates associated with the short and small crack data may seem alarming from an engineering perspective. However, SCC activity will only occur on-load and that means 16 h per day and 5 days per week, ignoring any crack retardation associated with the off-load period. Assuming 35 ppm chloride ion and a growth rate of $4 \times 10^{-10}$ m/s, a crack extension of 23 μm/cycle would be expected, which is equivalent to approximately 0.46 mm per month of excursion, whether continuous or discontinuous. Of course, the excursion in the concentration of chloride ion could be greater or smaller in magnitude and only service data coupled with additional testing would yield insight. What the analysis does highlight is the possible criticality of this mode of cracking, not so much that it would ultimately lead to failure directly, but in combination with low-frequency corrosion fatigue crack growth in normal water chemistry, the crack size could evolve to a depth where the threshold for high cycle fatigue cracking is exceeded. Clearly, the emphasis has to be on limiting chemistry excursions and restoring to normal water chemistry as expeditiously as possible.

### 6. Concluding Remarks

Quantifying the very early stages of precursor and crack development remains a major challenge because of the impact of surface condition and its variable nature both in service and in laboratory testing, fabrication-induced defects, and uncertainty over the impact of environmental excursions in operating conditions. This challenge applies also to cracks developed from corrosion pits for which the very small crack size as the micro-crack develops inherently limits reliable crack growth rate measurement. There is greater confidence in quantifying the growth of cracks in the small crack regime, when the crack front extends beyond the pit and in providing a framework for estimating time to the crack defection limit and remanent life. To limit the range of testing, modelling is essential. Models of crack electrochemistry provide qualitative scientific insight and can predict trends, with many examples of successful application. However, the extension of such models to quantitative crack growth rate prediction accounting for crack-tip mechanics, crack-tip electrochemical kinetics on bare metal, material microstructure and varied modes of crack advance is still a formidable challenge.

**Funding:** This research was funded by the National Measurement System of the United Kingdom Department for Business, Energy, and Industrial Strategy.

**Institutional Review Board Statement:** Not applicable.

**Informed Consent Statement:** Not applicable.

**Data Availability Statement:** Not applicable.

**Conflicts of Interest:** The authors declare no conflict of interest. The funders had no role in the design of the study; in the collection, analyses, or interpretation of data; in the writing of the manuscript, or in the decision to publish the result.

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
