# Peer review of "Reflections on Early Stages of Environmentally Assisted Cracking from Corrosion Pits"

_cmd, doi:10.3390/cmd2040030_

Round 1

Reviewer 1 Report

Interesting review, it provides a good analysis of the early stages of environmentally assisted cracking that occurs in stainless steels. It would have been more interesting to go further on the possible implication of the hydrogen coming from the cathodic activity of the pits, and also in the description of the implication of the metallurgy or microstructure at different scales (Ex-grain, grain boundaries, precipitates...).

On the other hand, I find that there is a lot of self-citation from the author, it would be important to cite other work from the community even if it is on other types of stainless steels or equivalent.

Author Response

I appreciate the suggestions on microstructure and hydrogen but I was keen to limit the scope to issues of measurement methodology as the range of possible factors if discussed in detail would be too ambitious in context.

The paper was not designed as a comprehensive review. That would have been enormous. That is why I used the term “reflections“ in the title and “brief overview” in the Introduction. It was more a commentary on the factors affecting the different stages of damage development using examples of our own work to illustrate. I have inserted a couple of sentences into the end of the Introduction.

Reviewer 2 Report

Crack initiation is an important and challenging topic in the environmentally assisted cracking (EAC) community. This review provides some insight into the early stage of crack initiation. However, some issues need to be addressed.

1, the title of this paper should be narrowed down as it only covers a small branch of EAC initiation.

2, I feel this paper is not well logically structured. It is better that the author describes how is this paper structured at the end of introduction.  Plus, part 5 seems not well connected to the previous part as it focus on steam turbine blades while the previous parts are mainly about the roles of pitting.

3, Some mistakes can be found in the text. For example, page6, line168, "digital image correction" seems to be "digital image correlation" . I will not list all of them, please check carefully.

Author Response

  1. Title changed as requested
  2. The paper is about the early stages of damage development that includes crack growth - end of Introduction changed to clarify objective
  3. Example of mis-type corrected but my attempt at proof reading failed to find others!

Reviewer 3 Report

I found this an interesting and clear review of the current understanding of early stage corrosion pit cracking.  

My only suggestion is that it would benefit from a more explicit indication in the title or abstract that it only really dealt with pitting related crack pre-cursors rather than all types of EAC related pre-cursors

Author Response

Title changed